# Single-Cell Genomics: Enabling the Functional Elucidation of Infectious Diseases in Multi-Cell Genomes

**DOI:** 10.3390/pathogens10111467

**Published:** 2021-11-12

**Authors:** Shweta Sahni, Partha Chattopadhyay, Kriti Khare, Rajesh Pandey

**Affiliations:** 1INtegrative GENomics of HOst-PathogEn (INGEN-HOPE) Laboratory, CSIR-Institute of Genomics and Integrative Biology (CSIR-IGIB), Mall Road, Delhi 110007, India; shweta.sahni@igib.in (S.S.); partha.c@igib.in (P.C.); kriti.29khare@gmail.com (K.K.); 2Academy of Scientific and Innovative Research (AcSIR), Ghaziabad 201002, India

**Keywords:** Single Cell Genomics (SCG), NGS, cellular heterogeneity, infectious diseases, immune response, host-pathogen interaction, AI

## Abstract

Since the time when detection of gene expression in single cells by microarrays to the Next Generation Sequencing (NGS) enabled Single Cell Genomics (SCG), it has played a pivotal role to understand and elucidate the functional role of cellular heterogeneity. Along this journey to becoming a key player in the capture of the individuality of cells, SCG overcame many milestones, including scale, speed, sensitivity and sample costs (4S). There have been many important experimental and computational innovations in the efficient analysis and interpretation of SCG data. The increasing role of AI in SCG data analysis has further enhanced its applicability in building models for clinical intervention. Furthermore, SCG has been instrumental in the delineation of the role of cellular heterogeneity in specific diseases, including cancer and infectious diseases. The understanding of the role of differential immune responses in driving coronavirus disease-2019 (COVID-19) disease severity and clinical outcomes has been greatly aided by SCG. With many variants of concern (VOC) in sight, it would be of great importance to further understand the immune response specificity *vis-a-vis* the immune cell repertoire, the identification of novel cell types, and antibody response. Given the potential of SCG to play an integral part in the multi-omics approach to the study of the host–pathogen interaction and its outcomes, our review attempts to highlight its strengths, its implications for infectious disease biology, and its current limitations. We conclude that the application of SCG would be a critical step towards future pandemic preparedness.

## 1. Introduction

The remarkable complexity of the biological processes involved in development, physiological homeostasis, disease, and infection outcomes is governed by the heterogeneity of cell states, cell fates, and cell types. An unravelling of these dynamics was brought about by the development of technological toolkits that allowed the detection and quantification of differential gene expression. The traditional methods of gene-expression profiling involve microarrays and bulk RNA Sequencing (RNA-Seq) using Next Generation Sequencing (NGS) methods, which broadly measure average gene expression levels in heterogeneous tissue [1]. The advent of single-cell RNA sequencing (scRNA-Seq) has led to unprecedented insights into cell composition and patterns of gene expression. Single cell transcriptomics has yielded new understanding of the dynamic cellular processes, such as development and differentiation [2,3,4]. The ever-growing body of scientific evidence obtained from scRNA-Seq studies has enabled the identification of newer cell types/subtypes and rare subpopulations [5,6]. Single-cell technologies have also profoundly enhanced our comprehension of cellular plasticity, transcriptional dynamics, and gene regulatory relationships [7,8]. Figure 1 represents the role of scRNA-Seq in understanding cellular heterogeneity through unicellular transcription and translation dynamics. Since the publication of the first scRNA-Seq transcriptomes in 2009 by Tang et al., the field of single-cell transcriptomics has grown tremendously in scope and magnitude, aided by the application of high throughput technologies, such as automation and microfluidics [9].

Studies of single cells have been traditionally performed in biological and medical fields through the use of microscopy, flow cytometry, and in situ hybridisation experiments. The advent of NGS has revolutionised single-cell studies by enabling the simultaneous profiling of the whole transcriptome (RNA-seq) of up to tens of thousands cells at once [10,11]. For all the advantages associated with bulk RNA-seq experiments, however, there is one limitation: they average out the gene expression levels across heterogeneous tissues [12]. They are thus unable to capture the expression pattern from individual cells. Single-cell transcriptomics, on the other hand, captures the stochasticity of gene expression in a heterogeneous population of cells [13]. Table 1 compares and contrasts the peculiarities of bulk and single-cell RNA sequencing. Heterogeneity among cell types and tissue composition may significantly impact the discovery of differentially expressed genes or the expression of quantitative trait loci (eQTLs) [14]. Consequently, the design of experiments is gradually moving towards an unbiased, hypothesis-free approach, and towards the analysis of tissues and organs in their entirety, rather than focusing on specific cell types. This has enabled pioneering discoveries with respect to profiling the changes in composition of different tissues associated with diseases such as diabetes, cancer or pathogen infection/s [15,16,17]. Therefore, scRNA-Seq has found extensive application in both basic and translational research.

## 2. Different Platforms for scRNA-Seq: The Quest for the Optimal Platform

Over the past decade, scRNA-Seq technologies have enabled a precise and unbiased view of the molecular mechanism at the single-cell resolution. Since the first scRNA-Seq method was published in 2009 [9], many other advancements in scRNA-Seq technologies have been developed. Early attempts at profiling single cells involved the sorting of individual cells into single wells of a plate followed by lysis, cDNA generation, barcoding, library generation, pooling, and sequencing [18,19,20]. Conventional scRNA-Seq procedures entailed the manual isolation of cells using techniques such as micromanipulation, Laser Capture Microdissection (LCM) or Fluorescence Activated Cell Sorting (FACS) [21,22,23]. However, the individual sorting of cells prior to downstream processing allows the assessment of cell viability, although it is strenuous, resource-intensive, and time-consuming. Advancements in cellular barcoding and microfluidics have allowed the upscaling and automation of single-cell isolation, thereby reducing the number of operational steps and leading to improvements in throughput [24,25,26]. Microfluidics-based scRNA-Seq platforms allow relatively cost-effective and large-scale parallel sequencing of thousands of cells at one time, while also reducing input reaction volumes from microlitres to nanolitres [27]. Integrated Fluidic Circuits (IFC), Droplet encapsulation or nanowell entrapment are employed on microfluidic platforms for capturing single cells.

The IFC-based automated array solution in the form of the Fluidigm C1 system was the first microfluidics system [28,29]. Single-cell suspensions are passed through fluidic circuits, wherein individual cells are immobilised in hydrodynamic traps, lysed, and further processed in nanoliter reaction chambers. However, IFC-based systems are limited by bias in sampling complex sample sources due to the array format, which is restricted to specific cell sizes. Further advancements in microfluidics led to the development of open nanowell-based systems, which allow higher scalability in terms of the cell numbers that can be sampled. Seq-well is a nanowell-based method that works on the principle of preloading the reaction wells with barcoded beads before the cells enter the wells [30]. The Seq-well system offers the advantages of portability, flexibility, and cost-effectiveness. However, a major disadvantage of the Seq-well system is that it does not allow the integration of imaging data due to the random distribution of barcoded beads in wells. In a similar way to the Seq-well, the BD Rhapsody system uses a planar array of nanowells for the capture of single cells, where nanowells are preloaded with barcoded beads before the cells are dispensed into them. The array format of nanowells offers various benefits, such as ease of operation, the direct imaging of cells to exclude doublets, the determination of cell viability, and the ability to examine cell phenotypes. In a nanowell-based approach, cell lysis occurs within the nanowells, allowing the indexing of mRNA transcripts on the barcoded magnetic beads, which is then followed by the pooling of barcoded beads for cDNA amplification and library preparation.

Despite their scalability to higher throughput and their easy operation, IFC- and nanowell-based approaches exhibit drawbacks for a limited number of reaction sites. However, on droplet-based microfluidic platforms, such as the single-cell Chromium controller from *10X Genomics*, individual cells are encapsulated in nanoliter droplets containing reaction reagents, thereby overcoming this challenge [31,32]. The number of sampled cells increases in linear progression with higher emulsion volume, facilitating large-scale scRNA-Seq studies. Cell lysis then proceeds within the beads and is followed by library preparation and sequencing. The advantages of droplet-based methods include reduced requirements for input reagents and samples, a lower number of operational steps, a higher throughput, increased scalability, and improved sensitivity and specificity. However, since barcodes are randomly introduced into droplets in droplet-based systems, they preclude the possibility of associating barcodes with images and the visual detection of cell properties.

### 2.1. Specialisations in scRNA-Seq Methods

In 2011, Islam et al. enhanced barcode-labelling technology for the first time and developed a highly multiplexed scRNA-Seq method called single-cell tagged reverse transcription sequencing (STRT-Seq) [18]. The strengths of STRT-Seq include its ability to pinpoint the exact location of the 5′ end of transcripts that could be used to analyze promoter usage in single cells. Subsequently, Hashimshony et al. proposed an advanced scRNA seq protocol, CEL-Seq (Cell Expression by Linear amplification and Sequencing), in which mRNA samples are barcoded and pooled before linear amplification by in-vitro transcription [33]. Thus, CEL-seq produced more reproducible, linear, and sensitive results than a PCR-based amplification method. Later, Jaitin et al. introduced an automated, massively parallel single-cell RNA sequencing (MARS-Seq) approach in which single cells from the target population were FACS-sorted to explore cellular heterogeneity within the immune system [34]. Subsequently, a new development in the field of scRNA occurred when Macosko et al. introduced Drop-seq, a method for the analysis of mRNA expression in thousands of individual cells by separating them into nanoliter-sized aqueous droplets for parallel analysis [35]. Buenrostro et al. developed a single-cell assay for transposase-accessible chromatin using sequencing (scATAC-seq), which utilises prokaryotic Tn5 transposase and a programmable microfluidics platform for mapping the accessible genomes of individual cells and to provide insights into cell-to-cell variation [36]. Furthermore, scATAC-seq can be integrated with scRNA-seq in order to perform multi-omic studies [37].

Datlinger et al. invented CROP-seq by combining CRISPR screening with single-cell RNA sequencing, which aids in the high-throughput functional analysis of complex regulatory mechanisms and heterogeneous cell populations [38]. In 2017, Vitak et al. proposed a single-cell combinatorial marker sequencing technique (SCI-seq), which is capable of simultaneously constructing thousands of single-cell libraries and detecting somatic copy-number variants [39]. In the same year, Chen et al. developed Linear Amplification via Transposon Insertion (LIANTI), a single-cell, whole-genome amplification method that can detect CNV at kilobase resolution and more effectively detect mutations in more diseases [40]. Guo et al. developed single-cell multiplex/multi-omics sequencing, or scCOOL-seq (Chromatin Overall Omic-scale Landscape Sequencing), which allowed the simultaneous analysis of single-cell chromatin accessibility, nucleosome positioning, copy number variations, ploidy, and DNA methylation with high sensitivity and coverage [41]. Habib et al. developed a single-cell nuclear RNA sequencing method by combining single-nucleus RNA-seq with microfluidic technology (DroNc-seq) for highly sensitive and efficient cell sorting, in which a variety of cells can be analyzed accurately [42]. Another high-throughput method for scRNA-seq was developed by Lake et al., through the integration of single-nucleus droplet-based sequencing (snDrop-seq) and single-cell transposome hypersensitive site sequencing (scTHS-seq), which is capable of detecting nuclear transcripts and epigenetic features to analyze gene expression and regulation in complex tissues [43]. Casasent et al. developed Topographic Single-Cell Sequencing (TSCS), which provides the accurate spatial location of individual tumor cells and assists in studying the invasive tumor cell populations [44]. A high-throughput and low-cost scRNA-seq technology, Microwell-seq, developed by Han et al., allows the improved detection of cellular heterogeneity and the characterization of cross-tissue cellular networks at the single-cell level [45]. Rosenberg et al. [24] developed split-pool, ligation-based transcriptome sequencing (SPLiT-seq), a low-cost, scRNA-seq method that performs comprehensive single-cell transcriptome sequencing through combinatorial barcoding, thus enabling flexible and scalable cell and sample multiplexing. Damaree et al. [46] developed single-cell genomic sequencing (SIC-seq), a high-throughput and low-bias method for scRNA-seq that utilises droplet microfluidics to isolate, amplify, and barcode the genomes of single cells. A combination of CRISPRi and scRNA-seq methods was developed by Gasperini et al. [47] to study and analyze the function of regulatory elements along with the interrelationship between regulatory elements and genes.

The frequent updates and development of scRNA-Seq technologies help single-cell genomics studies to discover the variability and dynamics of single-cell populations and their role in disease severity and clinical outcomes.

### 2.2. Experimental Design for scRNA-Seq: Plausible Challenges

As depicted in Figure 2, the general workflow of any single-cell experiment includes sample preparation, which involves the isolation of single cells, cell lysis, and mRNA capture, followed by single-cell RNA sequencing, data analysis, and data processing. The sequencing of single cells involves two main challenges not associated with bulk RNA-seq procedures: (1) the isolation and capture of individual cells, and (2) the adequate amplification of the miniscule amounts of mRNA obtained from single cells. The first stage involves achieving a single-cell suspension free from dead cells, cellular aggregates, and debris. This is the most critical and challenging step in a single-cell sequencing experiment. It is at this stage that specific cell types can be enriched or eliminated. This is followed by the capture of single cells within droplets or nanowells, and lysis within individual wells or droplets. To create sequencing-ready libraries, poly(A) tailed mRNA transcripts are captured using poly(T) oligonucleotides with Unique Molecular Identifier (UMI) sequences and single-cell-specific barcodes, which are converted to cDNA through a reverse transcription step. The oligonucleotides are designed to contain adaptors or T7 polymerase promoter sequences in order to allow the subsequent amplification of cDNA by PCR or in-vitro transcription (IVT) [18,33,48]. The amplified cDNA is then fragmented via either enzymatic fragmentation or mechanical forces in order to obtain sequencing-ready fragments. This is followed by a final amplification that attaches sequencing adaptors to the amplicons. The scRNA-Seq protocols may employ either full-length transcript sequencing or 3′-end/5′-end transcript sequencing technologies [18,35,49,50].

One of the first questions to address while setting up a single-cell experiment is the number of cells required to obtain data with the requisite strength for downstream analysis. Further, for the efficient demultiplexing and segregation of the single-cell data, it is crucial to ensure proper cell and sample barcoding throughout the workflow. In any single-cell transcriptomics study it is important to minimise the impact of stress-induced transcriptional changes during sample processing [51]. The obscurity of transcriptional profiles due to manual sample handling will eventually be overcome as progress is made towards automated and standardised tissue handling. Every single-cell experiment requires the user to make informed choices about such questions as the sample selection, the optimal cell number, the preparation platform, the choice of scRNA-Seq techniques, the selection of sequencing parameters, and the computational analysis strategies to obtain useful insights from scRNA-Seq data. Emerging computational tools allow the mitigation of batch effects by resolving the impact of biological and technical variation [52,53]. Generalised designs do not work optimally for single-cell transcriptomic studies and each one needs appropriate customisation at different stages, in line with the research question addressed.

## 3. Analysis of scRNA-Seq Data

The number of tools for analysing scRNA-Seq data is increasing steadily, each of them with its own advantages and disadvantages [54,55]. The low capture efficiency, high dropout rate, higher technical noise and higher biological variability (such as stochastic transcription) compared to conventional bulk RNASeq data impose substantial challenges on the computational analysis of scRNA-Seq data [56]. Although a variety of analytical tools is available for bulk RNA-Seq data analysis, most of them cannot be applied directly to scRNA-Seq data analysis [54]. A certain disparity exists between the analyses of bulk and scRNA-Seq data, such as differential gene expressions, gene regulatory networks, and cell clustering. Unlike whole-transcript sequencing, analyses aimed towards the identification of alternate splicing, allele-specific expression, and RNA editing events, are not suitable for 3′ or 5′ transcript sequencing protocols. Figure 3 represents the basic scRNA-Seq data analysis pipeline.

### 3.1. Quality Control of scRNA-Seq Data

After sequencing, the data from low-quality cells are eliminated, primarily based on the low number of reads. While FastQC, the commonly used quality control tool for bulk RNASeq data, can serve the purpose, there are some scRNA-Seq-data-specific QC tools, such as Scater [57], and SinQC [58], which offer better quality control. The mapping ratio of the reads to the reference is an important quality indicator for RNASeq data. The most commonly used mapping tools, such as HISAT [59], TopHat2 [60], and STAR [61], map billions of reads (both scRNA-Seq and bulk RNASeq) to a reference with great accuracy and speed. However, for the quantitation of gene expression, the conventionally used techniques, i.e., de novo assembly and reference-based assembly, can be used for single-cell whole-transcript sequencing, but not for 5′ or 3′ end-transcript sequencing. This quantitation is possible either using scRNA-Seq specific tools, such as Smart-seq2 and MATQ-seq, or tools for bulk RNASeq data analysis, such as Cufflinks [62], RSEM [63], and Stringtie [64]. For 5′ or 3′ end-transcript sequencing, UMI-based analysis pipelines, such as SAVER, offer an accurate estimation of the transcript count with very low technical noise [65].

Disparities in capture efficiency, sequencing depth, dropout reads, and noise between samples may result in data bias. The normalisation of scRNA-Seq data is therefore important for downstream analysis, such as the identification of cell subpopulations and differential gene expressions. Within sample normalization, the adjustment of the GC content and the transcript length allows better analysis of gene expression within one sample. By contrast, the between-sample normalisation of the sequencing depth and dropout enables comparison of gene expression across samples [66]. The normalization of scRNA-Seq data using conventional RNASeq tools, such as DESeq2 [67] and TMM [68], requires that specific precautions are taken, as higher technical noise and abundant zero expression values may result in overcorrection.

The dropout event increases the chance of cell-to-cell variability, which may affect the gene expression and gene–gene relationship analysis. To take this into account, imputation replaces the missing data with substitute values. Commonly used imputation tools include SAVER [65], ScImpute [69], MAGIC [70], AutoImpute [71], and DrImpute [72].

### 3.2. Analysis of Cellular Heterogeneity

As scRNA-Seq data are high-dimensional, it is important to reduce their dimensionality. Dimensionality reduction, along with feature selection, allows better visualisation and delineation of cells into subpopulations. Tools/methods such as Principal Component Analysis (PCA), t-Distributed Stochastic Neighbor Embedding (t-SNE), Uniform Manifold Approximation and Projection (UMAP), and scvis usually project the high-dimensional data onto a lower dimensional space, while preserving the key properties of the original data [24,73,74,75]. The unveiling of cellular heterogeneity and the identification of cell subpopulations is a major goal of single-cell RNA sequencing. The clustering of subpopulations is either based on known markers or de novo identification (unsupervised clustering methods, such as hierarchical clustering, k-means, graph-based clustering, and density-based clustering) [76]. Some clustering tools, such as SC3 and Seurat, clusters cell subpopulations based on statistical analysis of the expression of known markers and differential gene expressions [77,78]. Concurrently, cell trajectory analysis from scRNA-Seq data facilitates the identification of the cues for cell state transition.

### 3.3. Differential Gene Expression Analysis

Differential expression of genes (DEG) analysis is extremely important in order to identify the significantly differentially expressed genes and to interpret biological differences across subpopulations/samples/conditions. The high noise, dimensionality, and coexistence of multiple states of the same cell make the differential expression analysis a challenging one. Thus, the tools used for DEG analysis from bulk RNASeq data are not an optimal choice for scRNA-Seq data. Specific tools have been designed for this purpose; some of the most widely used are SCDE [79], BCseq [80], MAST [81], and Census [82].

### 3.4. Artificial Intelligence in scRNA-Seq Data

Until now, a huge amount of RNASeq data is available publicly. The data generated by single-cell RNA sequencing are huge and more complex than bulk RNA-Seq data. This requires specific analytical pipelines for scRNA-Seq data. With the implementation of Artificial Intelligence (AI)-based models, such as Machine-Learning (ML) and deep-learning methods (DL) in the analysis of genome sequencing data over the last 10 years, more accurate analysis of big data is now possible, without the need to model the system of interest. AI approaches are commonly used to solve regression, classification, dimensionality reduction, and clustering tasks. AI algorithms can be employed to capture more detailed information on cell types, DEGs, biomarker expression patterns, lineage transition, and disease subtypes, as well as to predict clinical outcomes [83]. AI-enabled analysis of scRNA-Seq data, along with the visualisation of landmark genes, enables us to uncover the “where” for every “what”, and offers a holistic understanding of gene expression at a single-cell resolution within a tissue microenvironment.

## 4. Balance between Expected Experimental Outcome, Cost, and Technology Preparedness

The fields of biological discovery and biomedicine have been greatly empowered by the recent technological advancements brought about in single-cell technologies. However, the relatively high costs and associated technical challenges have, so far, limited the widespread application of single-cell sequencing technologies. A concerted effort in the field has been aimed at ameliorating the cost-effectiveness of single-cell technologies and simplifying the experimental challenges involved. The advent of new sequencing technologies, the miniaturisation of reaction volumes, and the associated reagent usage have considerably reduced the costs of single-cell sequencing [84]. These robust advances and cost savings in tandem have enabled the growing footprint and application of single-cell sequencing in disease biology and basic research.

Beside its cost, there are a number of other technical and experimental challenges that must be overcome in order to facilitate the greater usage of scRNA-Seq in addition to that of bulk RNA-seq. The enormous amplification of the minute amounts of starting material from a single cell, when combined with scarce sampling, leads to consequential bias [85]. The distortion in gene expression profiles is also, in some cases, a result of the low RNA capture efficiency and conversion rate [86]. The aggregate effect of all this is that gene expression profiling by scRNA-Seq is inherently noisier than the bulk RNA-seq datasets. Further, significant distortion in expression profiles also results from drop-out events due to the occasional failure to detect the transcripts otherwise expressed at a high level in the cell, leading to false inter-cellular variability [87]. The sub-optimal experimental design, in terms of processing samples at different times, may be another source of technical variation due to batch effect, significantly affecting the results of scRNA-Seq. Besides, barcode impurities or the external background often cause errors with demultiplexing during the performance of data analysis. Taken together, these subtle, yet significant, technical details limit the data calibration and interpretation of scRNA-Seq experiments. Apart from the technical sources of variation, significant biological variation also poses challenge to data reproducibility in single-cell experiments. The use of spike-in and the dilution of bulk RNA to single-cell levels are some of the approaches currently used for the evaluation of the nature and magnitude of technical variability; however, each has its own limitations [88]. We are sure that the future development of more targeted approaches to the delineation of the technical and biological variability of single-cell gene expression profiling data would take into account the above challenges.

## 5. Single-Cell Sequencing in Infectious Disease

Single-cell sequencing has brought about a revolution in our understanding of the complex systems biology of the immune landscape. The human immune system is composed of a large number of cell types and states, which together regulate the pathophysiology of many diseases, such as cancer, autoinflammatory disorders, and infectious diseases. However, a majority of the population of the immune system requires further characterization. The cells of the immune system, i.e., T lymphocytes, B lymphocytes, Natural Killer cells (NK Cells), and macrophages respond to pathogen challenges and release a wide variety of cytokines, antibodies, and complement proteins in order to clear infections. At the same time, they express cell surface markers, such as a wide variety of clusters of differentiation (CD), B cell receptors (BCR), T cell receptors (TCR), the Major Histocompatibility Complex (MHC), and the Human Leukocyte Antigen-DR isotype (HLA-DR). Apart from these, the number of specific cell types and subtypes also changes during the immune response. Thus, understanding the complexities of the immune system and the spatio-temporal expression of the genes, surface markers, and cell types of the immune system during an infection may help to understand diseases better and significantly improve their management. Figure 4 depicts the use of scRNA-Seq in understanding infectious disease biology, wherein the single-cell RNA sequencing of infected samples helps to understand the host response and the host–pathogen interactome.

### 5.1. Studying the Immune Atlas

#### 5.1.1. Immune Cell Repertoire Profiling

The immune repertoire refers to the B cell and T cell population with distinct antigen specificity at any given time. The receptor on the B cell surface (BCR), which primarily binds to the antigen, is composed of two heavy (H) and two light (L) chains. The heavy chain is further composed of a variable region (V), a diversity region (D), a joining region (J) and a constant region (C), while the light chain is composed of V, J and C regions only. T cell surface receptors (TCR) are composed of α and β chains, and somatic V(D)J recombination leads to tremendous diversity within an individual TCR repertoire. Any two T cells bearing the same TCRαβ sequences are likely to arise from the same ancestor; therefore, identifying the BCR and TCR sequences can reveal the ancestry of T cells, which is especially important when the antigen is unknown [89]. Single-cell V(D)J sequencing was enabled by scRNA-Seq, revealing the BCR and TCR sequences; this accelerated the identification of neutralising antibodies against invading pathogens. This has been useful to the understanding of the immune response during the current COVID-19 pandemic, as has been highlighted by multiple studies. Using scRNA-Seq-mediated BCR sequencing, Xi et al. identified 14 potent neutralising antibodies against SARS-CoV-2 [90]. Using single-cell TCR sequencing, Liao et al. reported a substantial increase in CD8+ clonally expanded cells in mild SARS-CoV-2 infections, but not in severe infections [91]. The V(D)J combination of BCR is unique to a pathogen. For instance, the V(D)J combination observed during SARS-CoV-2 infection is different from the V(D)J combination during other infections [92]. Using BCR and TCR sequencing, Schultheiß et al. reported the lowering of T cells and an increase in B cell numbers in patients with active COVID-19 infections. They also reported a shift of CD4+:CD8+ T cell ratios towards CD4+ T cells [93]. Stephenson et al. reported the expansion of proliferating lymphocytes and monocytes, platelets and hematopoietic stem and progenitor cells as disease severity increased [94]. They also reported a decrease in IgA+ cells in patients with symptomatic COVID-19 infection compared to asymptomatic patients, despite the expansion of the plasma cell population. The depletion was mainly mediated by depleted IgA2 subtypes. However, in contrast to Schultheiß et al., Stephenson et al. reported an expansion of CD8+ T cells. Such variation may be taken as evidence for diversity in immune responses during COVID-19 infection. Thus, BCR and TCR sequencing using scRNA-Seq provide important insights into the adaptive immune response and accelerate the identification of antibodies, which is essential for the development of vaccines.

#### 5.1.2. Identification of Novel Cell Subtypes

During pathogenic challenges, a compendium of heterogenous host immune cells is involved in important biological processes, such as pathogen recognition, neutralising antigen, and antigen presentation. Transcriptional changes and/or changes in the expression of surface markers may occur during infections, either as a result of infection, or in order to clear the infection. The identification of novel immune cell subpopulations, and an understanding of their characteristics, are important for understanding infection dynamics. Waickman et al. reported a group of clonal expanded CD8+ T cells with unique transcriptional characteristics [95]. CD4+ T cells differentiated into two types during latent HIV infection, with lower host and viral transcription levels in type 1 cells [96]. Using Seq-Well, Geirahn et al. reported novel subtypes of macrophage cells during Mtb infections [30]. Martin-Gayo et al. reported three types of conventional dendritic cells (cDCs) in HIV patients; one of the three subsets showed adistinct transcript expression profile and a high antiviral response, which may have elicited a better immunological response [97]. While most cellular heterogeneity analysis depends on the similar transcript expression profile of individual cells, at the same time, it is important to take necessary precautions while clustering cell subpopulations in order to avoid false reports of novel subtypes.

#### 5.1.3. Immune Signalling Pathways

The scRNA-Seq method provides differential gene expression information at a single-cell resolution, which makes accurate mapping of the signalling pathway of a cell possible. Horns et al. reported a fraction of peripheral memory B cell activation after administering an influenza vaccine, while a fraction of memory B cells were not activated. They reported 172 differentially expressed genes across the two groups, revealing the signalling pathways in the memory B cells [98]. Wen et al. reported a higher expression of genes related to inflammation-associated signalling pathways in NK cells in COVID-19 patients [99]. Inflammatory factors perform various biological roles in the regulation of the immune system. Using scRNA-Seq, Zhang et al. reported the spatio-temporal production of inflammatory factors in IAV-infected lungs [100]. He et al. highlighted that the production of IL-1β and TNF-α upon SARS-CoV-2 infection may lead to extra mucin secretion, which may contribute to Acute Respiratory Distress Syndrome (ARDS) [101]. These findings highlight that the use of scRNA-Seq enables us to understand the immune signalling pathways at a single-cell resolution.

### 5.2. Understanding the Host–Pathogen Interaction

#### 5.2.1. Host–Pathogen Diversity

Understanding the pathophysiology of an infectious disease requires a detailed understanding of the host cells, as well as the pathogen. A variety of evidence shows that a change in host-cell diversity occurs in response to a pathogen challenge. How does the infection affect the host cell’s diversity? How does the host cell’s diversity affect the infection outcome? In the case of multiple cell types infected by the same pathogen, do the cells share similar features that make them susceptible to the infection? Are all the single-pathogen bodies infecting specific cell types the same? Answers to these fundamental questions are critical in order to understand the pathophysiology of infection.

The analysis of pathogen-infected cells requires segregation between infected and healthy cells. Xin et al. investigated the effect of host cell heterogeneity on infection and found that host cell size was a regulator of infection [102]. A 10-100-fold increase in viral titre was reported in cells in the G2/M phase [103]. The scRNA-Seq method provides an unprecedented resolution for the analysis of host cellular heterogeneity during an infection. Multiple studies have elucidated the transcriptional landscape of the immune system during infection. This provides a clearer picture of the host’s response to the pathogen. A scRNA-Seq of COVID-19 infected patients showed a marked increase in CD4+ and CD8+ T cells and plasma B cells in BALF and PBMCs. Megakaryocytes and CD14+ monocytes were elevated during the early and severe infection stages of COVID-19, while NK cells, γδ T cells, cDCs, plasmacytoid dendritic cells (pDC), and CD16+ monocytes were depleted. The depletion of NK cells, pDCs, and CD16+ monocytes was correlated with COVID-19 associated with severe breathing distress [104,105]. Park et al. reported the transformation of NK cells into innate lymphoid-cell-1-(ICL1)-like cells, capable of producing IFN-γ but not TNF-α, upon *Toxoplasma gondii* infection [106].

Pathogen diversity may be inherent, or it may arise as a result of the host–pathogen interaction. As most current scRNA-Seq technologies use oligo dT to capture transcripts, positive strand RNA viruses, having poly A tail, are also captured and can be detected in deep sequencing. Negative strand RNA viruses can also be detected in scRNA-Seq analysis by using specific probes to capture the viral transcript. Multiple studies reported a diverse range of viral loads and intracellular viral RNA in cells infected with IAV, even though all the parameters were kept the same throughout each study [103,107]. Russellet al. reported that IAV is prone to mutation during infection [108]. Although there is substantial evidence for virus diversity during infection, bacterial diversity during infection at the single cell level is poorly understood.

#### 5.2.2. Infection Dynamics

Understanding the dynamics of infection enables us to understand the proliferation and promulgation of pathogens in vivo and their role in pathogenesis. Ramos et al. analysed the IAV-respiratory epithelial cell interaction dynamics during the early stage of infection. They reported that a high multiplicity of infection (MOI) of IAV leads to a high intracellular viral mRNA, which suppresses the host’s innate immune response in a similar way to the suppression of IFN production [109]. Zanini et al. identified the flavivirus infection-associated host factors involved in endoplasmic translocon, membrane trafficking, and signal peptide processing, by studying the flavivirus–host-cell interaction dynamics using scRNA-Seq [110]. A study showed that in-silico TCR reconstruction, combined with the transcriptome sequencing of T cells, led to the mapping of T cell activation dynamics during Salmonella infection [111]. Using a scRNA-Seq of nasal swabs from COVID-19 patients, Qi et al. reported that ACE2, TMPRSS2, NRP1, and NRP2 were more expressed in the nasal epithelial region of symptomatic COVID-19 patients than in asymptomatic patients. They also observed mild inflammation and enhanced epithelial barrier function, along with an increased CD8+ T cell response, in asymptomatic COVID-19 patients, compared with symptomatic patients, which may explain the absence of symptoms in a proportion of COVID-19 patients [112].

#### 5.2.3. Antibody Response

The interrogation of the antigen specificity of B cells in order to identify a correct B cell clone from thousands or millions of B cells is an important focus of research. Along with the use of oligo-barcoded antibodies, scRNA-Seq has made it easier to identify the correct B cell clone. El Debs et al. co-encapsulated single hybridoma cells, an enzyme ACE1, and its fluorescent substrate within water in oil microdroplets to identify the ACE1-inhibiting antibody [113]. The PBMCs of severe COVID-19 patients showed a higher amount of plasma B cells (~15%) compared to healthy patients and those with a lower degree of infection (~3%). Furthermore, these plasma B cells were enriched for genes encoding the constant regions of IgA1, IgA2, IgG1, and IgG2, suggesting their role in the secretion of antibodies against the infection [105]. Cao et al., using a high-throughput scRNA-Seq and V(D)J sequencing, identified a total of 14 potent SARS-CoV-2 neutralising antibodies from a pool of 8558 antigen-binding IgG1+ cells from antigen-enriched B cells. Out of the 14 antibodies, one antibody, named BD-368-2, was found to be very potent against SARS-CoV-2. The finding offers a promising therapeutic and prophylactic strategy [90].

#### 5.2.4. Identification of Susceptible Cells Subtypes

The identification of cell types susceptible to pathogens enables us to understand infection mechanisms with more clarity. Using scRNA-Seq and cell clustering analysis, it has been reported that the influenza virus primarily infects and colonises in the lung’s epithelial cells. A higher level of viral mRNA in the lung’s epithelial cell supported this finding [114]. The scRNA-Seq method has also been used to identify potential target cells for the SARS-CoV-2 infection. Using this technique, a high expression of ACE2, BSG, DPP4, and ANPEP, as well as the S protein proteases, TMPRSS2 and CTSL, are found in the placenta during the first trimester [115]. This indicates a higher risk of transmission to the foetus during pregnancy. Another study provided evidence of the expression of ACE2 and TMPRSS2 in the proximal convoluted tubule, the proximal tubule, the distal tubule, and the glomerular parietal epithelium of the kidney, which may explain SARS-CoV-2-induced kidney damage [116]. These results, based on scRNA-Seq analysis, identified key information about the entry and transmission of viruses.

## 6. Future Directions

The past few years have seen rigorous growth in the field of single-cell genomics, marked by the development of chemistries, methods, platforms, and techniques that simplify and economise both experiments and the analysis of single-cell sequencing. Single-cell sequencing offers enormous potential for the improvement of our understanding of biological problems and the fundamentals of human disease. Inter-disciplinary applications of scRNA-Seq have expanded the frontiers of various realms of biological research, such as neurology, oncology, immunology, and developmental and regenerative biology (depicted in Figure 5).

The profiling of the human body at single-cell resolution is expected to accelerate advancements in basic and clinical research, with translational benefits for future medicine and diagnostics. While individual efforts at the single-cell profiling of major human organs have long been underway, the recent emergence of international consortia aimed at deciphering single-cell transcriptomes is of great importance for future research [117]. The integration of single-cell sequencing data from these consortia is expected to lead to systematic data integration and harmonization, enabling the obtainment of the universal human cell reference dataset. The Human Cell Atlas is one such international collaborative initiative, aiming to bring together expertise in biology, genomics, medicine, and computation in order to create a concerted database of all the cells in the human body [118,119]. Various tissue- and disease-specific consortia, such as the BRAIN initiative, the LungMAP, and others have enabled focused elucidation of the intricacies of individual organs and their associated diseases [120,121]. Single-cell genomics technology is of overarching significance in bringing these ideas together.

As single-cell sequencing technology becomes more pervasive and ubiquitous, it is of paramount importance to simultaneously address the need for platforms that enable data handling of such enormous magnitudes. At the same time, the increasing repertoire of computational biology methods, in tandem with the utilisation of artificial intelligence, is expected to further empower scRNA-Seq to play an ever-growing role in disease diagnosis, treatment, and human welfare. Progress at multiple levels would contribute towards future pandemic preparedness.

## Figures and Tables

**Figure 1 pathogens-10-01467-f001:**
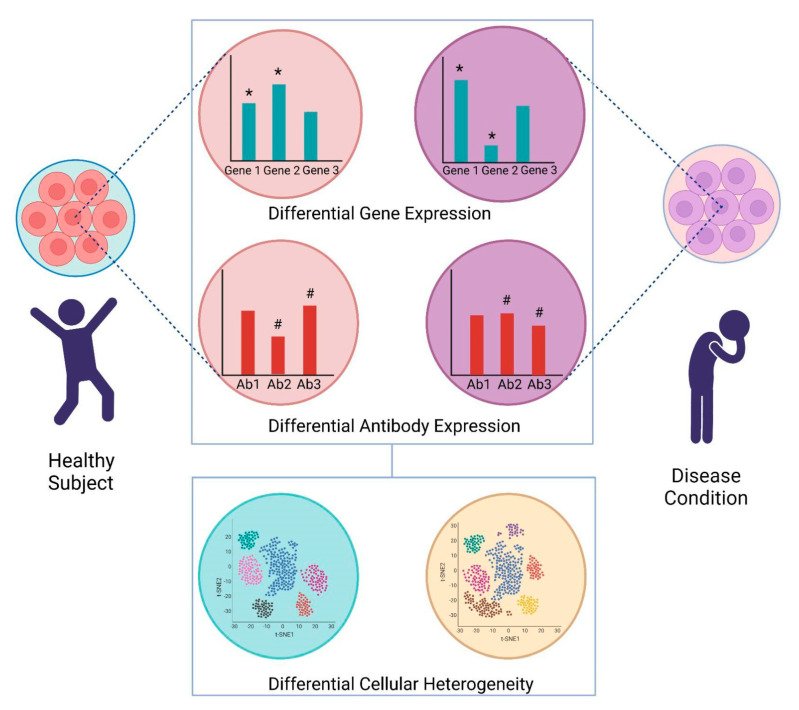
Graphical representation highlighting the importance of scRNA-Seq in bridging the gap between transcriptome and translation. The gene expression pattern of same cell from a healthy and a disease individual is different (highlighted as *), and the expression of antibodies/surface markers are also different (highlighted as #). However, the gene expression does not always correspond to the degree of the antibody/surface marker expression. Using complementary sequencing approaches, it is possible to undertake whole transcriptome amplification (WTA) for expression profiling at the single-cell level, as well as Antibody-seq (Ab-seq) for selected antibodies to capture the dynamics of transcriptome-translation.

**Figure 2 pathogens-10-01467-f002:**
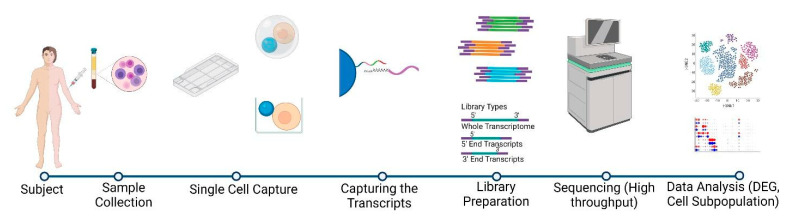
Basic steps involved in scRNA-sequencing. These include the production of single-cell suspension from isolated tissues/blood (PBMCs) while maintaining the individual cells identity, followed by multi-step single-cell sequencing-ready library preparation (whole transcript/5′ end/3′ end transcript), and high-throughput sequencing and analysis to capture the functional role of cellular heterogeneity.

**Figure 3 pathogens-10-01467-f003:**
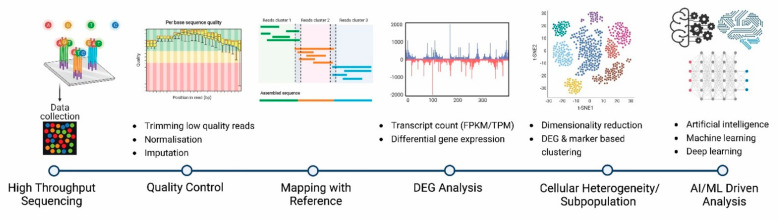
An overview of scRNA-Seq data analysis pipeline. The summary of the different stages of the analysis highlights the opportunities and challenges of the scRNA-Seq. It is important to mention that AI/ML has contributed to a better understanding of the scRNA-Seq data and its potential applications.

**Figure 4 pathogens-10-01467-f004:**
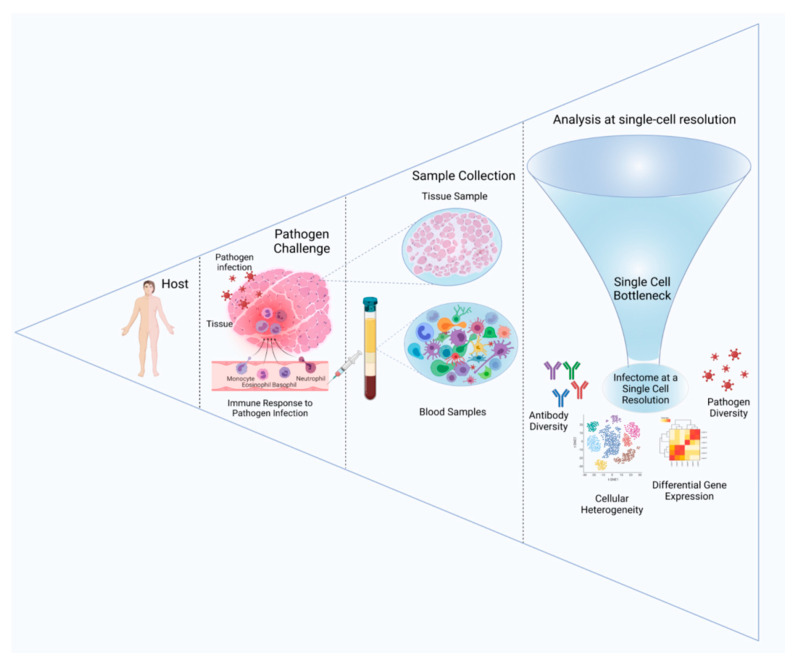
Landscape of Single-cell Genomics from the perspective of infectious disease biology. In response to pathogen infection in a host, samples can be taken from either tissue or blood. Subsequently, through a multi-step process involving single-cell library preparation, sequencing, and analysis, functional inferences are drawn. Interactomes at a single-cell resolution would help in the formation of inferences about antibody diversity, differential gene expression, cellular heterogeneity and pathogen diversity.

**Figure 5 pathogens-10-01467-f005:**
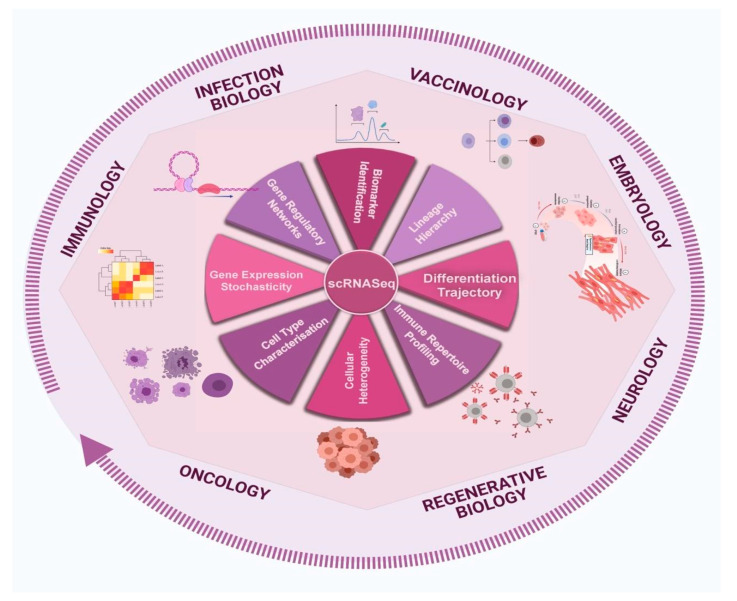
Diverse trajectories and applications of the scRNA-Seq. The figure captures some of the aspects wherein understanding from scRNA-Seq has increased knowledge across multiple fields, including immunity, neurology, vaccination, cancer, and regenerative biology. This has been made possible due to the single-resolution information from specific studies.

**Table 1 pathogens-10-01467-t001:** Comparison of bulk RNA-Seq and scRNA-Seq for gene expression studies.

		Bulk RNA-Seq	scRNA-Seq
**Objective**	Goal	1. Measure average gene expression across a population of cells. 2. Identify differences between sample conditions.	1. Measure gene expression of individual cells. 2. Identify differences between cell types.
	Protocol	1. RNA extraction, reverse transcription, fragmentation, adaptor ligation, amplification, and sequencing.	1. Single cell isolation, RNA extraction along with cell specific barcode labelling/UMI tagging, reverse transcription, adaptor ligation, amplification, and sequencing.
**Experimental design**	QC and analyses	1. No of genes/transcripts per sample. 2. Between sample normalisation. 3. Differential gene/transcript expression.	1. Number of genes/transcripts per cells. 2. Percentage of genes mapped to mitochondria. 3. Batch effect normalisation. 4. Imputation. 5. Immune phenotyping. 6. Cell subpopulation identification. 7. Differential gene expression.
**Technical considerations**	Specialised Instrumentation	1. Not required for making the sequencing ready library.	1. Specialised instrument for single cell separation required (e.g., BD Rhapsody, 10× Chromium Controller and Seqwell).
	Sample Prep	1. RNA integrity and quality needs to be ensured during RNA extraction.	1. Careful and gentle handling of cells required to ensure high viability and minimal cellular aggregate formation.
**Resource** **Intensivity**	Cost	1. Comparatively economical.	1. Higher per sample cost
	Expertise	1. Normal NGS expertise needed.	1. Cell handling, instrumentation and analyses require special expertise.
	Computational	1. Computationally less intensive.	1. Extensive computational infrastructure required. 2. Big data storage.

## Data Availability

Not applicable.

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
