# Peer review of "Single-Cell Genomics: Enabling the Functional Elucidation of Infectious Diseases in Multi-Cell Genomes"

_pathogens, 2021, doi:10.3390/pathogens10111467_

Round 1

Reviewer 1 Report

The review from Sahni et al was focused on single cell genomics and multi-cell genomics. This is an interesting topic, however, the review stay very much on the surface of the topic.

The first paragraphs deal more or less with the technique and the data analysis of SCS, but does not provide enough depth.

The authors come up with SGS and bulk sequencing, they should be more critical on this point. What are the advantages and disadvantages.

Also the role of SGS in multi-cell genomics and disease, is very superficial. The authors should be more critical on the role of SGS in the the analysis of immune phenotyping and disease development. In particular this is a very interesting topic and the authors should focus more on this aspect. The role of SGS and again a little bit more of a critical discussion.

All in all, this review is very much descriptive and should be more a discussion!

Author Response

Dear Reviewer,

We thank you for the peer review of the manuscript [pathogens-1278922].

The authors take this opportunity to acknowledge and thank the anonymous reviewer for contribution towards reviewing the manuscript in a detailed manner and sharing valuable inputs. We are sure that addressing the suggestions would help improve the quality of the manuscript in terms of readability and information content.

The detailed response to reviewer is attached for your perusal.

Best wishes,

Rajesh

Reviewer 2 Report

The title should somehow refer to the special 'infectious disease' focus of the manuscript.

Navin et al., 2011 is cited wrongly. There is no RNA-seq data in that paper. They investigated genomic DNA from single nuclei. The authors should read all papers and cite those properly.

Section 2.1. 'single cell genomic experiment'. Transcriptomic, not genomic since the focus of the paper is scRNA-seq. Genomics and transcriptomics should not be used interchangably throughout the paper.

Figures are not cited in the text.

Fig 2 is not a general scRNA-seq workflow as its title suggests. Neither from technology nor from sample point of view.

Section: In the first sentence, I wouldn't call it 'sequencing depth'. Rather 'low number of reads' or 'low number of detected genes/UMIs'.

bias instead of biasness

What does '100K of RNASeq data' mean?

Author Response

(The authors gave the same response as above.)

Round 2

Reviewer 1 Report

the authors improved the review and answered all my critical points.